# Economic system justification predicts muted emotional responses to inequality

Shahrzad Goudarzi[1], Ruthie Pliskin[2], John T. Jost [1] & Eric D. Knowles[1]*

Although humans display inequality aversion, many people appear to be untroubled by widespread economic disparities. We suggest that such indifference is partly attributable to a belief in the fairness of the capitalist system. Here we report six studies showing that economic ideology predicts self-reported and physiological responses to inequality. In Studies 1 and 2, participants who regard the economic system as justified, compared with those who do not, report feeling less negative emotion after watching videos depicting homelessness. In Studies 3–5, economic system justifiers exhibit low levels of negative affect, as indexed by activation of the corrugator supercilii muscle, and autonomic arousal, as indexed by skin conductance, while viewing people experiencing homelessness. In Study 6, which employs experience-sampling methodology, everyday exposure to rich and poor people elicits less negative emotion among system justifiers. These results provide the strongest evidence to date that system-justifying beliefs diminish aversion to inequality in economic contexts.

[1] Department of Psychology, New York University, 6 Washington Place, New York, NY 10003, USA. [2] Institute of Psychology, Leiden University, Wassenaarseweg 52, 2333AK Leiden, The Netherlands. *email: edk202@nyu.edu

Adults, children, and even some non-human primates respond negatively to the unequal distribution of valued resources—even when inequality is personally advantageous[1]. Children as young as 6 years prefer egalitarian resource allocations[2] and may refuse desired items to avoid having more than their peers[3]. Likewise, chimpanzees have been observed to discard high-value rewards (grapes) when conspecifics receive low-value rewards (cucumbers) for performing the same task[4]. Given this evidence of deep-seated inequality aversion, one might expect people, in general, to be highly concerned about the widening chasm between the rich and poor[5]. However, public opinion data suggest that a large percentage of Americans either pay little attention to or are otherwise unperturbed by widespread economic inequality[6–9].

System justification theory (SJT) seeks to identify the belief systems that blunt the aversive impact of societal inequalities[10]. One such belief system is economic system justification (ESJ)—the belief that the capitalist system provides individuals with equal opportunity to succeed, and that outcomes are based upon personal deservingness and merit[11]. This conviction, in turn, enables system justifiers to interpret patterns of wealth and poverty as fair, legitimate, and appropriate, thereby reducing distress in the face of inequality. Corroborating this theoretical account, political conservatives—known to be relatively high in ESJ—report greater happiness and life satisfaction than liberals, with the happiness gap increasing during periods of heightened economic inequality[12]. Moreover, ESJ reduces moral outrage, guilt, distress, and support for redistributive social policies[13]. Such findings suggest that belief in the legitimacy of the capitalist system serves a palliative function, shielding people from aversive experiences that are otherwise triggered by inequality[10]. Nevertheless, existing evidence for the ideological palliation thesis has been challenged in two significant ways.

Central to the ideological palliation thesis is the notion that economic ideology shapes affect when inequality is salient. However, research cited in support of ideological palliation[12,13] does not clearly isolate the relationship between economic beliefs and responses to inequality. Ideological differences in affect might reflect not only the power of system-justifying economic beliefs to lessen inequality aversion but rather the influence of other factors associated with system-justifying ideologies—such as insensitivity to a broad range of macro-level threats[14] or low levels of motivation to feel empathy[15]. By experimentally manipulating the salience of economic inequality and accounting for the effects of general empathy, the studies reported here allow us to focus on the link between system-justifying beliefs and reactions to inequality.

Critics of existing research on the palliative effects of system-justifying ideologies have also argued that ideological differences in self-reported happiness and positive affect reflect individual variation in self-enhancement tendencies rather than authentic emotional experiences[16]. Thus, it is possible that conservatives (typically higher in ESJ) present themselves as—but do not genuinely feel—happier than liberals (typically lower in ESJ)[17]. The present research circumvents this concern by leveraging physiological indices of negative affect and arousal, including measures of facial muscle contraction and electrodermal activity, in response to others' economic suffering.

Across six studies, we hypothesize and find that exposure to extreme manifestations of inequality (e.g., people experiencing homelessness) elicits less negative affect in people who regard the prevailing economic system as more (vs. less) fair and legitimate. In Studies 1 and 2, we assess discrete self-reported emotions pertaining to the homeless people and the US economic system as a whole. In Studies 3–5, we index negative affect via activation of the corrugator supercilii (brow) muscle and affective arousal via skin conductance levels (SCL). In Study 6, we assess emotional experiences following quotidian exposure to poor and wealthy individuals. Detailed hypotheses and data-exclusion criteria for Studies 2, 4, and 5 were preregistered on the Open Science Framework (OSF) and complete data from all studies are available from the OSF repository (https://osf.io/2qn2z). Informed consent was obtained from participants in all studies. Moreover, all studies reported in this paper complied with ethical regulations for work with human subjects and the study protocol was approved by NYU's University Committee on Activities Involving Human Subjects.

## Results

**Studies 1 and 2: self-reported affective experience.** Workers from two crowdsourcing marketplaces, Amazon Mechanical Turk[18] (Study 1; $N = 105$) and Prolific Academic[19] (Study 2; $N = 326$), took part in a two-session study. In the first session, participants completed Jost and Thompson's 17-item ESJ scale[11]. Sample items include "If people work hard, they almost always get what they want," "Economic positions are legitimate reflections of people's achievements," and "There are many reasons to think that the economic system is unfair" (reverse-scored). To assess the effects of social desirability concerns on affective reports[16,17], we administered the Balanced Inventory of Desirable Responding (BIDR)[20] to participants in Studies 1 and 2. To examine the effects of individual differences in generalized empathy, we administered the Empathic Concern (EC) subscale of the Interpersonal Reactivity Index[21] in Study 2.

Approximately 1 week after completing pretest measures, participants were invited to complete the second part of the study. The procedures of the second session differed slightly between the two experiments. In both, participants watched two video clips in randomized order. In the homeless condition, participants watched interviews with people experiencing homelessness in which they discussed their daily routines and struggles; any references to potentially incriminating behavior, such as theft or drug use, were edited out. In the control condition, participants viewed interviews on mundane topics. Study 2 included an additional video type—specifically, interviews with people suffering from cystic fibrosis (CF), an inherited disease. The CF video allowed us to explore whether the palliative effects of ESJ would extend to misfortunes that are not directly related to economic disadvantage.

After watching each clip, participants in Study 1 reported how much anger, sadness, disgust, and guilt they felt toward the individual in the video and the "American socioeconomic system." Ratings of pity and empathy for the homeless person, and hope for and pride in the system were also collected. In Study 2, only person-directed sadness, anger, and empathy and system-directed anger, sadness, and disgust were assessed. Supplementary Tables 1–4 and 23–28 present descriptive statistics, including bivariate correlations among emotions assessed in Studies 1 and 2. We hypothesized that individuals high (vs. low) in ESJ would report less negative emotion in response to the homeless videos (vs. control videos). Because the data were nested, with each participant providing affective reports for homeless and control videos, mixed-effects linear regressions were conducted to examine the effects of ESJ (z-scored), video type (1 = homeless, 0 = control), and the ESJ × Video Type interaction on affective reports. Intercepts varied randomly between participants and robust SEs were specified. We adjusted for video order in the analysis and stimuli were effect-coded within video type to adjust for within-condition heterogeneity between the videos.

Primary results of Study 1 are shown in Table 1; see Supplementary Tables 5–16 for full regression results and

**Table 1 Person- and system-directed emotions as a function of economic system justification (ESJ) and video type (homeless vs. control) in Study 1.**

|  | Video type | ESJ × Video type |
|---|---|---|
| Person-directed emotions |  |  |
| Sadness | 63.708 (2.841)*** | −9.274 (2.594)*** |
| Pity | 57.336 (2.774)*** | −10.729 (2.809)*** |
| Empathy | 53.708 (3.129)*** | −7.363 (3.061)* |
| Anger | 13.448 (2.221)*** | 0.747 (2.112) |
| Disgust | 7.528 (2.038)*** | 1.695 (1.823) |
| Guilt | 30.023 (2.876)*** | −3.008 (3.051) |
| System-directed emotions |  |  |
| Anger | 36.393 (2.924)*** | −10.448 (3.000)*** |
| Sadness | 45.074 (2.800)*** | −9.532 (2.922)*** |
| Disgust | 36.762 (3.086)*** | −13.338 (3.149)*** |
| Guilt | 26.475 (2.897)*** | −3.917 (2.934) |
| Pride | −18.327 (2.660)*** | −2.054 (2.817) |
| Hope | −10.390 (2.626)*** | −0.467 (2.928) |

***$p < 0.001$, *$p < 0.05$. Cells contain unstandardized coefficients and their SEs (in parentheses). Video type is coded such that 1 = homeless and 0 = control. Emotions were rated on a 0–100 scale

**Table 2 Person- and system-directed emotions as a function of economic system justification (ESJ), empathic concern (EC), and exposure to the homeless videos in Study 2.**

|  | Homeless video (H) | ESJ × H | EC × H |
|---|---|---|---|
| Person-directed emotions |  |  |  |
| Sadness | 67.197 (1.563)*** | −6.806 (1.693)*** | 9.919 (1.783)*** |
| Pity | 61.169 (1.715)*** | −3.245 (1.864)† | 7.497 (2.030)*** |
| Empathy | 53.935 (1.963)*** | −3.353 (2.138) | 4.786 (2.289)* |
| System-directed emotions |  |  |  |
| Anger | 45.673 (1.880)*** | −11.170 (2.010)*** | 3.875 (2.050)† |
| Sadness | 49.033 (1.871)*** | −7.374 (2.159)*** | 7.024 (2.013)*** |
| Disgust | 45.914 (1.916)*** | −11.548 (1.971)*** | 4.248 (2.033)* |

***$p < 0.001$, **$p < 0.01$, *$p < 0.05$, †$p < 0.1$. Cells contain unstandardized coefficients and SEs (in parentheses). Homeless video is coded such that 1 = homeless and 0 = control. Emotions were rated on a 0–100 scale

Supplementary Figs. 1–12 for graphs of the relevant interactions. Video type had a significant effect on all person- and system-directed emotions: participants reported more sadness, anger, disgust, and guilt toward the person and the socioeconomic system, more empathy and pity toward the person, and less pride and hope in the system after watching the homeless (vs. control) video. Moreover, we observed the hypothesized ESJ × Video Type interactions for sadness, pity, and empathy toward the person in the video and anger, sadness, and disgust toward the socioeconomic system. The negative signs of these interaction terms indicate that video type had a weaker effect on the emotions of participants scoring high (vs. low) in ESJ. After applying a Bonferroni correction to limit familywise error across the 12 interaction tests, the effects for person-directed sadness and pity, and system-directed anger, sadness, and disgust remained significant.

The results were robust to model specifications that included gender, race, age, religiosity, and income as covariates (Supplementary Tables 17–22). To examine the effects of self-presentational biases, we also tested a model that included the two BIDR subscales (Self-Deceptive Enhancement and Impression Management) in the regression equation along with their interactions with video type[22]. Results were similar whether BIDR scores were included in the model or not (Supplementary Tables 17–22).

Hypotheses of Study 2 were informed by the results of Study 1 and were preregistered on the OSF platform (see https://osf.io/2qn3z). We again hypothesized that video type (homeless vs. control) would have weaker effects on the emotional reactions of individuals who were high (vs. low) in ESJ. Furthermore, as CF is a primarily non-economic form of suffering, we hypothesized that there would be no difference in emotional responses to the CF (vs. control) video as a function of ESJ.

Following the preregistered analysis plan, we conducted mixed-effects linear regressions to examine the effects of ESJ (z-scored), the homeless video contrast (1 = homeless, 0 = control), the CF video contrast (1 = CF, 0 = control), and the ESJ × Homeless and ESJ × CF interactions on emotion reports. Intercepts varied randomly between participants and robust SEs were specified. We adjusted for video order in the analysis and stimuli were effect-coded within video type to adjust for within-condition heterogeneity between the videos. See

Supplementary Tables 29–34 for full regression results and Supplementary Figs. 13–18 for simplified interaction graphs.

Exposure to the homeless video significantly increased the strength of all person- and system-directed emotions. Moreover, the predicted ESJ × Homeless interactions were observed. High (vs. low) ESJ participants reported smaller increases in all negative emotions upon viewing the homeless (vs. control) videos (see Supplementary Information, Study 2 Supplementary Results). Contrary to expectations, we observed significant ESJ × CF interactions for person-directed sadness and system-directed anger, sadness, and disgust, such that participants low (vs. high) in ESJ reported more negative emotions in response to the person with CF. Importantly, however, the ESJ × CF interactions were significantly smaller than the ESJ × Homeless interactions for all assessed emotions (see Supplementary Information, Study 2 Supplementary Results). The results were robust to inclusion of the BIDR, gender, race, age, religiosity, and income as covariates (Supplementary Tables 35–40).

Previous research suggests that self-identified conservatives are less motivated to feel empathy than liberals[15]. Consistent with these findings, we found that ESJ and EC were negatively correlated ($r = −0.32$, $p < 0.001$). To ensure that our findings reflected the effects of ESJ rather than liberal–conservative differences in dispositional empathy, we adjusted for EC by including it and its interactions with the video contrasts in a series of regression models. Results are summarized in Tables 2 and 3. In these models, the ESJ × Homeless interactions for person-directed sadness and system-directed anger, sadness, and disgust remained significant with the inclusion of EC (Table 2). Of the ESJ × CF interactions, only those for system-directed anger and disgust survived the inclusion of EC in the model (Table 3). Simple effects analyses revealed that ESJ effects for all emotions were significantly larger in the homeless condition than in the CF condition (see Supplementary Information). The fact that ESJ moderated some of the emotional responses to CF sufferers (albeit weakly) suggests that low-ESJ participants may have attended to the potential economic harms of the disease or to shortcomings of the US healthcare system. Hence, for these participants, CF may not have represented an entirely non-economic form of suffering.

Research on ideological differences in affective responses has sometimes been framed in terms of general political orientation[12,17,23], which typically ranges from "very liberal" to "very conservative." From the perspective of SJT, conservatism should be associated with muted affective reactions to inequality in large part because conservatives are more likely to regard the economic system as fair and justified[24]. We therefore sought to

**Table 3 Person- and system-directed emotions as a function of economic system justification (ESJ), empathic concern (EC), and exposure to the cystic fibrosis videos in Study 2.**

|  | Cystic fibrosis video (CF) | ESJ × CF | EC × CF |
|---|---|---|---|
| Person-directed emotions |  |  |  |
| Sadness | 70.166 (1.526)*** | −0.333 (1.649) | 12.561 (1.593)*** |
| Pity | 59.425 (1.783)*** | 0.258 (1.909) | 7.741 (2.087)* |
| Empathy | 56.716 (1.924)*** | −1.122 (2.064) | 6.622 (2.067)** |
| System-directed emotions |  |  |  |
| Anger | 23.231 (1.786)*** | −4.410 (1.764)* | 4.477 (1.920)* |
| Sadness | 32.307 (1.826)*** | −2.721 (1.934) | 6.992 (1.938)* |
| Disgust | 22.968 (1.706)*** | −4.983 (1.710)** | 3.409 (1.957)† |

***$p < 0.001$, **$p < 0.01$, *$p < 0.05$, †$p < 0.1$. Cells contain unstandardized coefficients and SEs (in parentheses). Cystic fibrosis video is coded such that 1 = cystic fibrosis and 0 = control. Emotions rated on a 0–100 scale

**Table 4 Indirect effects of conservatism on inequality treatment effects mediated by economic system justification in Study 1.**

|  | B | 95% CI |
|---|---|---|
| Person-directed emotions |  |  |
| Sadness | −1.870 | **[−3.491, −0.815]** |
| Pity | −1.823 | **[−3.311, −0.846]** |
| Empathy | −1.604 | **[−3.284, −0.160]** |
| System-directed emotions |  |  |
| Anger | −1.692 | **[−3.327, −0.507]** |
| Sadness | −1.649 | **[−3.107, −0.555]** |
| Disgust | −1.697 | **[−3.328, −0.073]** |

Bootstrapped and accelerated confidence intervals that exclude 0 are bolded

**Table 5 Indirect effects of conservatism on inequality treatment effects mediated by economic system justification in Study 2.**

|  | B | 95% CI |
|---|---|---|
| Person-directed emotions |  |  |
| Sadness | −2.336 | **[−3.514, −1.278]** |
| Pity | −1.476 | **[−2.540, −0.387]** |
| Empathy | −0.957 | [−2.111, 0.162] |
| System-directed emotions |  |  |
| Anger | −2.827 | **[−3.998, −1.634]** |
| Sadness | −2.212 | **[−3.399, −0.974]** |
| Disgust | −3.260 | **[−4.387, −2.196]** |

Bootstrapped and accelerated confidence intervals that exclude 0 are bolded

determine whether ESJ could account for previously observed relationships between general ideological self-placement and affective outcomes[12,17].

To prepare the data for analysis, we first subtracted emotion ratings in the control condition from their corresponding ratings in the homeless condition in each study; this difference score reflects the inequality treatment effect (homeless vs. control) for each participant's emotional responses. We then conducted a series of mediation analyses[25] to test the indirect effects—through ESJ—of general political orientation on the treatment effect for each emotion yielding a significant ESJ × Homeless interaction; bias-corrected and accelerated bootstrap confidence intervals were obtained on the basis of 1000 replicates. We adjusted for the same covariates included in the previously reported regression models in each study. In the case of the person- and system-directed emotions for which we observed ESJ × Video type interactions (with the exception of empathy in Study 2), the confidence intervals for the indirect effects excluded 0—suggesting that ESJ conveyed the influence of political orientation on these emotions (see Tables 4 and 5, and Supplementary Figs. 19–30). These findings support the notion that self-reported political orientation predicts emotional responses to inequality, and that this relationship is attributable to conservatives' stronger tendency to justify the economic system.

Studies 1 and 2 provide evidence that system-justifying beliefs attenuate the experience of negative emotions during encounters with economic inequality. Moreover, two features of the data imply that ESJ is uniquely related to affective responses to inequality. First, ESJ predicted reactions to economic suffering more strongly than it did responses to suffering of a non-economic nature (CF). Second, the effect of ESJ on emotional reactions to economic suffering was not reducible to differences in EC.

Recent critiques suggest that ideology and emotional self-reports may be confounded due to a reluctance among conservatives to admit experiencing negative emotion[16,17]. On this view, political conservatives may report—but not genuinely experience—less negative emotion than liberals. The fact that the results of Studies 1 and 2 were unaffected by the inclusion of a measure of social desirability concerns calls such critiques into question. Nonetheless, the next three studies further address criticisms of self-report methodologies by replicating the results of Studies 1 and 2 using physiological measures of affective experience.

**Studies 3–5: physiological indicators of affective experience.** In Studies 3–5, a total of 155 New York University undergraduates watched the video clips used in Study 1 while undergoing physiological measurement. Several weeks prior to the lab sessions, participants completed the ESJ scale. During the lab sessions, facial electromyography[26] was used to assess activation of the corrugator supercilii (brow) muscle as an index of negative affect[27,28]. Activation of the levator labii superioris muscle was also assessed (see Supplementary Information for a detailed description of the levator measure and the rationale for its inclusion). Muscle activity was measured in microvolts relative to a baseline reading taken prior to the start of each video[29]. Affective arousal was indexed by skin conductance level (SCL), which provides a reliable measure of autonomic nervous system arousal[30]. SCL was measured in microsiemens relative to a pre-stimulus baseline. Self-reported affective valence was assessed by having participants continuously rate how negatively or positively they felt during the video clips by sliding their forefinger left or right across a touchpad. We predicted that, consistent with the results of Study 1, homeless videos would induce lower corrugator activity, SCL, and self-reported negative affect among participants who scored high (vs. low) in ESJ.

Studies 3–5 were methodologically similar to one another, with three exceptions: (1) in Study 4 we lacked a measure of SCL; (2) Study 5 included the CF target type used in Study 2; and (3) in Study 3, the control condition always preceded the homeless condition, whereas the order of the video clips was counterbalanced in Studies 4 and 5 (see Supplementary Information for detailed descriptions of each experiment). Mixed-effects linear regressions were conducted to test the interactive effects of ESJ and video type on participants' corrugator activity, SCL, and self-reported affect. All models specified random intercepts and slopes for video type, unstructured covariance matrices, and robust SEs. Individual stimuli were effect-coded within each video type to

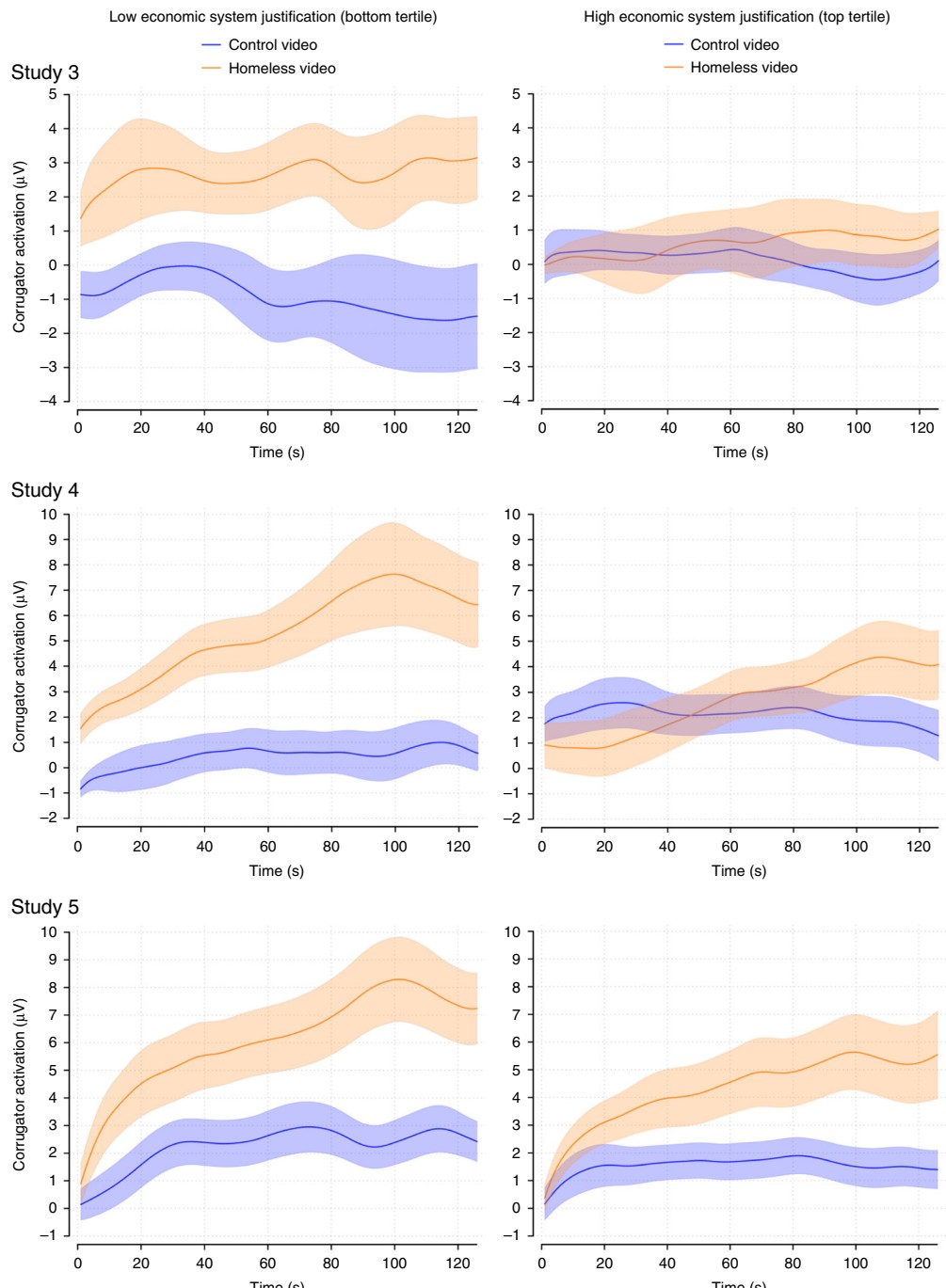

**Fig. 1 Economic system justification predicts muted corrugator supercilii activity during depictions of people experiencing homelessness.** The y-axis reflects activation relative to baseline. Colored bands indicate ±1 SE. LOWESS smoothing applied.

adjust for within-condition heterogeneity between the videos. In Studies 4 and 5, we also adjusted for stimulus order. See Supplementary Tables 41–58 for descriptive statistics and full regression results for Studies 3–5.

Figures 1a and 2a display descriptive results for corrugator and SCL in Study 3. Regression analyses yielded interactions such that the homeless video induced significantly weaker corrugator activity and SCL among participants high (vs. low) in ESJ. In Study 4 the corrugator effect was successfully replicated (Fig. 1b) and in Study 5 both the corrugator and SCL findings were replicated (Figs. 1c and 2b). In Study 5, as expected, the ESJ × CF interactions were nonsignificant—providing evidence that the palliative effects of ESJ, when measured at the physiological level,

did not extend to suffering of a primarily non-economic sort. Despite marginally significant interactions between ESJ and video type on levator activation and self-reported affect in Study 3, these trends were not replicated in Studies 4 or 5. The results of all physiological experiments are summarized in Table 6. See Supplementary Figs. 31–41 for simplified interaction graphs depicting the results of Studies 3–5.

We conducted an integrative data analysis (IDA) to examine the overall pattern of results across the physiological experiments (Studies 3–5), while accounting for between-study heterogeneity[31]. The results of this analysis are also summarized in Table 6 (see Supplementary Tables 59–62 for full regression results.) The IDA confirms that, across studies, exposure to homeless videos

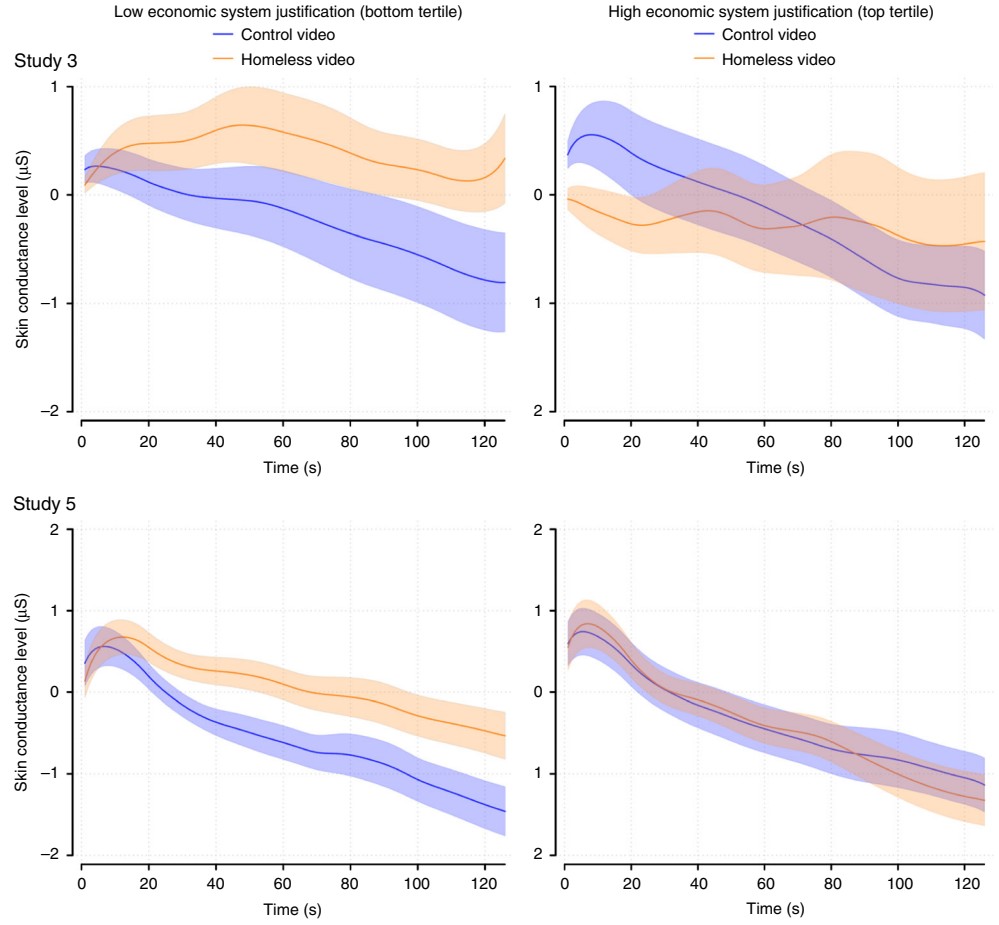

**Fig. 2 Economic system justification predicts reduced skin conductance levels during depictions of people experiencing homelessness.** The y-axis reflects activation relative to baseline. Colored bands indicate ±1 SE. LOWESS smoothing applied.

**Table 6 Psychophysiological responses as a function of economic system justification (ESJ) and video type in Studies 3–5 and IDA.**

|  | Corrugator activity (µV) | Levator activity (µV) | Skin conductance level (µS) | Positive affect |
|---|---|---|---|---|
| **Study 3 (N = 42)** |  |  |  |  |
| Homeless Video (H) | 1.368 (0.671)* | 0.781 (0.549) | 0.399 (0.197)* | −15.157 (2.841)*** |
| ESJ × H | −1.507 (0.736)* | −2.458 (−1.572) | −0.567 (0.256)* | 4.693 (2.390)† |
| **Study 4 (N = 37)** |  |  |  |  |
| Homeless Video (H) | 3.895 (0.853)*** | 1.645 (0.655)* | – | −9.656 (3.579)** |
| ESJ × H | −1.735 (0.742)* | −0.281 (0.631) | – | −0.176 (4.442) |
| **Study 5 (N = 76)** |  |  |  |  |
| Homeless Video (H) | 3.245 (0.438)*** | 0.893 (0.503)† | 0.543 (0.171)** | −7.407 (1.352)*** |
| ESJ × H | −0.896 (0.486)† | 1.171 (1.132) | −0.449 (0.211)* | 0.239 (1.639) |
| **Integrative data analysis (N = 155)** |  |  |  |  |
| Homeless Video (H) | 2.831 (0.389)*** | 1.077 (0.347)** | 0.406 (0.123)** | −10.328 (1.615)*** |
| ESJ × H | −1.192 (0.355)*** | −0.651 (0.588) | −0.472 (0.145)** | 0.431 (1.769) |

***$p < 0.001$, **$p < 0.01$, *$p < 0.5$, †$p < 0.1$. Cells contain unstandardized coefficients and their SEs in parentheses. Homeless video is coded such that 1 = homeless and 0 = control

elicited weaker corrugator and SCL responses among participants who were high (vs. low) in ESJ (see Supplementary Figs. 42–45 for simplified interaction graphs). These results were robust to model specifications that included gender, race, age, religiosity, and parental income as covariates (see Supplementary Tables 63 and 64).

As in Studies 1 and 2, we examined whether ESJ could account for previously observed relationships between general political orientation and affective outcomes in the integrated physiological dataset. Consistent with predictions preregistered on the OSF platform for Study 2 (see https://osf.io/2qn2z), mediation analyses showed that ESJ conveyed effects of political orientation on inequality treatment effects (homeless vs. control) for corrugator activity and SCL (see Table 7 and Supplementary Figs. 46 and 47). These findings provide additional support for the notion that political orientation predicts emotional responses to inequality through the association between conservatism and ESJ.

In Studies 1–5, we operationalized exposure to inequality through depictions of homelessness, but in their daily lives people may encounter inequality in many other forms—including

various manifestations of poverty and conspicuous wealth. Our final study addressed this possibility directly by asking people to record their day-to-day experiences with economic inequality, thus allowing us to conduct a more naturalistic investigation of the ideological palliation thesis.

**Study 6: affective responses to daily encounters with rich and poor people.** We used experience-sampling methods[32] to investigate associations between ESJ and affective responses to contexts of economic inequality in daily life. Following a quasi-experimental design, NYU undergraduates received four text messages a day for 9 consecutive days, prompting them to complete a short survey using their smartphones. Two of the daily surveys were designed to measure reactions to inequality, asking participants to indicate whether they had encountered someone they considered either very poor or very rich compared with themselves; each administration of the survey focused on one of these two categories, with the order randomized across days. Regardless of whether participants reported such an encounter, they were asked about their emotions—either in light of the encounter (if one was reported) or over the preceding two hours (if no encounter was reported).

ESJ was measured at the beginning of the academic semester as part of the NYU psychology battery. Consistent with the preceding experiments, we hypothesized that following encounters with poor people, low (vs. high) ESJ respondents would report feeling more anger and disgust at the system and more empathy and sadness toward the target. We also hypothesized that following encounters with rich people, low (vs. high) ESJ respondents would report feeling more anger and disgust at the system. For exploratory purposes, we also measured anger, jealousy, and empathy directed at rich individuals, as well as sadness about respondents' own socioeconomic status (see Supplementary Information, Study 6 Measures and Materials). Daily reports were nested within participants. Therefore, we conducted mixed-effects linear regression analyses to estimate the between- and within-subject effects of exposure to rich and poor

people in daily emotion reports—as well as the degree to which these emotional responses were moderated by ESJ[33]. We specified a random intercept and random slope of exposure to economic inequality for each participant.

Within-subject effects of encounters with inequality, as well their interactions with ESJ, are summarized in Table 8 (see Supplementary Tables 65–66 for descriptive statistics, Supplementary Tables 67–77 for full regression results, and Supplementary Figs. 48–58 for graphs of the relevant interactions). As hypothesized, we observed a significant and negative ESJ × Poor interaction for anger and disgust at the system. Analysis of simple effects revealed that low system justifiers displayed more system-directed anger and disgust upon encountering poor people than did their high-ESJ counterparts. We also observed an unexpected ESJ × Poor interaction for target-directed anger, such that participants low (but not high) in ESJ expressed anger toward the poor person. Despite being unpredicted, this result is broadly consistent with the ideological palliation thesis insofar as it reveals that high system justifiers reported lower levels of negative affect when confronted with inequality.

Turning now to daily encounters with rich people, we observed a significant and negative ESJ × Rich interaction for anger and disgust at the system. Analysis of simple effects revealed that low (vs. high) system justifiers displayed more system-directed anger and disgust upon encountering rich people. Consistent with the ideological palliation thesis, we also observed significant and negative ESJ × Rich interactions for target-directed anger and jealousy as well as sadness about one's own socioeconomic status. That is, individuals who were low (vs. high) in ESJ felt more anger, jealousy, and sadness when confronted with a rich person. These results were robust to inclusion of the BIDR, gender, race, age, religiosity, and income as covariates (Supplementary Tables 78–85).

Mirroring Studies 1–5, we next tested whether ESJ conveyed an effect of self-reported political orientation on emotional reactions to everyday experiences with inequality. First, for each participant, we subtracted mean levels of each emotion across days (i.e., instances) when the participant encountered inequality from the mean levels of emotions across days when they did not. This difference score reflects the inequality treatment effect (having in an encounter with inequality vs. not) for each participant's emotions. We then conducted a series of mediation analyses to test the indirect effects—through ESJ—of general political orientation on the treatment effect for each emotion; bias-corrected and accelerated bootstrap confidence intervals were obtained on the basis of 1000 replicates. The confidence intervals for the indirect effects of political conservatism on person-targeted and system-directed disgust after poor encounters, as well as target-directed anger and system-targeted anger and disgust in the case of rich encounters excluded 0, suggesting that ESJ conveyed an influence of political orientation on these

---

**Table 7 Indirect effects of conservatism on inequality treatment effects as mediated by economic system justification (ESJ) in the integrated physiological dataset.**

| Affect index | *B* | 95% CI |
|---|---|---|
| Corrugator activity | −0.509 | **[−1.136, −0.112]** |
| Levator activity | −0.227 | [−1.538, 0.417] |
| Skin conductance level | −0.201 | **[−0.439, −0.067]** |
| Self-reported valence | −0.643 | [−2.338, 0.909] |

Bootstrapped and accelerated confidence intervals that exclude 0 are bolded

---

**Table 8 Daily emotions as a function of ESJ and encounters with rich and poor people in Study 6.**

| | Poor | ESJ × Poor | Rich | ESJ × Rich |
|---|---|---|---|---|
| Empathy (target) | 1.979 (0.186)*** | −0.016 (0.185) | 0.282 (0.177) | 0.104 (0.174) |
| Anger (target) | 0.114 (0.112) | −0.302 (0.111)** | 0.418 (0.180)* | −0.408 (0.176)* |
| Sadness (target) | 2.043 (0.191)*** | −0.026 (0.190) | – | – |
| Jealousy (target) | – | – | 0.631 (0.185)** | −0.445 (0.181)* |
| Anger (system) | 1.092 (0.152)*** | −0.361 (0.152)* | 0.549 (0.188)** | −0.488 (0.185)** |
| Disgust (system) | 1.069 (0.149)** | −0.482 (0.148)** | 0.502 (0.195)* | −0.415 (0.191)* |
| Sadness (self) | – | – | 0.188 (0.150) | −0.335 (0.147)* |

***p < 0.001, **p < 0.01, *p < 0.05. Cells contain unstandardized coefficients and their SEs (in parentheses). Poor and rich encounters were coded such that 1 = encounter and 0 = no encounter. Cells corresponding to unmeasured emotions are left empty

**Table 9 Indirect effects of conservatism on inequality treatment effects mediated by economic system justification in Study 6.**

|  | *B* | 95% CI |
|---|---|---|
| **Poor encounter** |  |  |
| Anger (target) | −0.155 | **[−0.318, −0.008]** |
| Anger (system) | −0.077 | [−0.250, 0.096] |
| Disgust (system) | −0.173 | **[−0.332, −0.013]** |
| **Rich encounter** |  |  |
| Anger (target) | −0.326 | **[−0.582, −0.0689]** |
| Sadness (self) | 0.025 | [−0.174, 0.225] |
| Jealousy | −0.020 | [−0.336, 0.297] |
| Anger (system) | −0.300 | **[−0.586, −0.0133]** |
| Disgust (system) | −0.308 | **[−0.600, −0.0160]** |

Bootstrapped and accelerated confidence intervals that exclude 0 are bolded

emotions (see Table 9 and Supplementary Figs. 59–63). Consistent with previous studies, we again found support for the notion that self-reported political orientation predicts emotional responses to inequality to the extent that self-identified conservatives have a stronger faith in the legitimacy of the economic system.

## Discussion

The studies reported here suggest that ESJ may buffer individuals from experiencing negative emotions that would otherwise be triggered by exposure to economic suffering and inequality. Importantly, ESJ predicted psychophysiological responses more strongly following exposure to economic (vs. non-economic) forms of distress, and the effects of ESJ were not attributable to ideological differences in generalized EC. Insofar as psychophysiological measures are not subject to response biases that are known to affect self-report measures, we can be confident that high-ESJ participants did not merely report—but genuinely experienced—lower levels of emotional distress following exposure to homeless targets in Studies 1–5 (cf. ref. [17]).

Our findings should contribute to a more nuanced view of ideological differences in negativity bias. While previous work suggests that conservatives may be more reactive than liberals to negative (e.g., threatening and disgusting) stimuli[34,35], our work suggests that—when it comes to economic inequality and suffering—conservatives may be less emotionally reactive than liberals because of differences in ESJ. Given that homelessness may induce disgust in some observers[36], and that conservatives are generally more sensitive to disgust[34], the fact that low (vs. high) ESJ respondents were more emotionally responsive to homelessness may be slightly counterintuitive. However, we suggest that in situations of economic inequality the palliative effects of ESJ may be strong enough to counteract more general ideological differences in negativity bias. Future research would do well to explore these ideas further.

In the self-report studies we adopted a discrete emotion approach, because exposure to manifestations of inequality may elicit different negative emotions based on core appraisals associated with them and their target (e.g., the system vs. individuals). Our predictions, nonetheless, focused on how ideology shapes overall differences in experienced negativity, as modulated by economic ideology, when encountering inequality. These differential affective experiences could, in turn, relate to broader ideological differences in subjective well-being[12] as in the broaden-and-build theory of emotions[37], according to which emotional valence is linked to subjective well-being. Future research could expand on this by tying ideological differences in

emotional experiences to longer term subjective well-being and identifying discrete emotions that are involved in this process.

The conviction that the prevailing economic system is fair, legitimate, and justified is associated with reduced inequality aversion, thus providing evidence for the ideological palliation thesis (see also ref. [38]). There may be more than one way in which system-justifying beliefs palliate negative hedonic experiences associated with economic suffering and inequality. One possibility is that dispositional attributions for poverty and wealth are overlearned and spontaneous; this would enable system justifiers to avoid even the momentary experience of negative affect. Alternatively, there may be ideological differences in emotion regulation processes[39], such that people who are high (vs. low) in ESJ are more effective at downregulating their negative emotional reactions to inequality. It is also possible that people who are low in ESJ upregulate their negative emotional reactions to find fault —and justify moral indignation—with the economic system. We close by highlighting one potentially unsettling implication of our findings, namely that people may incur a psychological cost by refusing or otherwise failing to justify the economic system. If so, a formidable emotional hurdle may obstruct the path to adopting a more critical stance toward the economic system and promoting social justice and economic change.

**Reporting summary.** Further information on research design is available in the Nature Research Reporting Summary linked to this article.

## Data availability

The data that support the findings of this paper are available on the OSF repository (https://osf.io/2qn2z). A reporting summary for this Article is available as a Supplementary Information file.

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

## Acknowledgements

J.T.J. acknowledges support from National Science Foundation Award #BCS–1627691 and E.D.K. acknowledges support from National Science Foundation Award #BCS–1729295. We thank Ruchita De, Angela Wang, Kellianne Holland, Ehimamiegho Idahosa-Erese, Liesel Staubitz, and Casey McMahon for their devotion and diligence in assisting with this research. S.G. thank Koohyar Hosseini and Zeinab Shahidi for editing the video stimuli used in this research.

## Author contributions

S.G., E.D.K. and J.T.J. planned Studies 1–5. All authors planned Study 6. S.G. and E.D.K. prepared the study materials and conducted Studies 1–5. S.G. and R.P. prepared the study materials and conducted Study 6. S.G. and E.D.K. analyzed the data for all studies and all authors were involved in their interpretation. All authors wrote the manuscript and provided feedback at different stages of the research, reviewed, and approved the manuscript.

## Competing interests

The authors declare no competing interests.
