## [Peer Review File · Nature Communications]

Reviewers' Comments:

Reviewer #1:

Remarks to the Author:

Across five experiments the authors test the ideological palliation thesis using three different samples, and a mix of self-report and physiological outcome measures. They experimentally manipulate salience of inequality and control for a large number of alternative explanations in testing the hypothesized relationship between ESJ and affective responses to individuals suffering as a result of said inequality. The studies are more than competently executed, the results are consistent and robust, and the manuscript is very clearly written. I also want to commend the authors for pre-registering their predictions and making their data available. Finally: THAT TITLE!!! If it isn't already clear, I really enjoyed reading this paper and I think it makes an important contribution to our understanding of how ESJ shapes micro-reactions to macro factors. Below I outline some thoughts for the authors to consider as they revise the manuscript.

My only general concern is that the corrugator and SCL results consistently hovered around .03-.05 but the pre-registration plus the IDA significantly attenuate this concern.

Specific comments (in order of appearance in the paper):

p. 3: The discussion jumps from relationships between SJ and fairness to conservatives and life satisfaction/happiness. I understand that L/R and SJ are related but they're not the same so it makes it harder to follow the logic. It's also worth noting that the absence of distress (as predicted by the IPT) does not necessarily require or indicate the presence of high satisfaction/happiness. As such, it might just be cleaner to review literature regarding relative absence of negative affect among high SJs in the intro.

I don't see the EC scale in the robustness check regressions for Experiment 2.

Just to make sure I'm understanding Tables 2 and 3 correctly, each of the interaction columns is that interaction with the main effect of the other scale in the model, correct? E.g., the ESJ X H column indicates the predicted interactions on person-directed and system-directed outcomes treating EC as a covariate?

I think Table 6 has the experiments labeled incorrectly. Should be experiments 3-5 + IDA (not 2-4 + IDA).

I have an alternative hypothesis for why the continuous affect data didn't replicate Experiments 1 and 2, which is foreshadowed in my first specific comment above: the continuous measure captured Ps levels of POSITIVE affect (which they registered by sliding a finger left or right on a touchpad). But if you're watching videos about homelessness and CF there ought not be ANY positive affect. If I were a P, I would have kept the dial set at 0 (any other value would have reflected schadenfreude). Ideally Experiments 3-5 would have asked about intensity of negative affect instead.

Supp Mat:

Looks like there is missing text and punctuation here: "Contrary to our preregistered hypothesis, however, we did observe a significant, although smaller, ESJ X CF Video interaction effect on person-directed sadness, system-directed anger, sadness and disgust was marginally significant ESJ X CF Video interaction effect on person pity. Notably, the effect of ESJ X Video Type was significantly larger (with some as twice bigger) for the ESJ X Homeless interaction contrast as compared to the ESJ X CF

condition on person-directed sadness and system-directed anger, sadness, and disgust (see Tables S29–S34).

Mina Cikara

Reviewer #2:

Remarks to the Author:

In five studies, the current article examines whether economic system justification (esj) reduces negative affect in response economic suffering (homelessness).

After reviewing the results, for each study, I wondered whether esj was negatively correlated with negative affect in the homelessness condition. That is, whereas the authors report that the esj x condition interaction was generally significant, and the simple effect of condition was generally stronger at 1sd below the mean in esj than at 1sd above, it was unclear whether the simple slope describing the esj-affect relation in the homelessness condition was significant, as one would expect on the theory that esj “blunts the negative emotional impact of inequality”.

In addition, I wondered whether the data could be better characterized as showing that low esj heightens negative affect in response to economic suffering, rather than high esj blunting negative affect. This appears to be the case for the result depicted in Figure 1, where individuals higher in esj showed just as much negative affect as all participants did in response to someone who suffered from cystic fibrosis (i.e., even at high esj, participants were well above the midpoint in expressing negative emotion in the homelessness condition, far higher than negative emotion experienced in response to the control video, and on par with negative emotion in response to cystic fibrosis). Individuals low in esj expressed even more negative emotion in response to the homeless target (compared to higher esj participants). Thus, at least for this study, it doesn't appear that high esj really blunted negative affect, as much as low esj elevated it (if we take the cystic fibrosis video as a reference point for the experience of negative affect). It would be helpful to have a similar figure for the other studies to help understand if the pattern shown in figure 1 is consistent.

A more minor empirical note: given such high reliabilities for the person and system composites (e.g., table 3), it seems more parsimonious to report results with composites alone in the main text, perhaps leaving results with individual components to supplemental materials for interested readers.

On a theoretical level, I wondered whether these data are as consistent with social dominance theory as they are with system justification theory. To be clear, I completely agree that these data are consistent with system justification theory – that is, it is plausible that some combination of epistemic, existential, and relational motives would lead to general and economic system justification, which in turn could reduce negative affect in response to poverty. It seems plausible in addition, or alternatively, that individuals who are favorable to social stratification would likewise justify the current economic system, with consequences for affective responses to poverty. An exploratory test of this could be done with the data from experiment 3, where social dominance orientation (sdo) was measured. That is, just as analyses were done to examine if esj mediated the effect of conservatism on affect, one could examine if general system justification (also measured in exp. 3) and sdo simultaneously predict esj, controlling for one another, with downstream effects on affect.

Reviewer #3:

Remarks to the Author:

Goudarzi, Knowles and Jost present a series of studies to test a hypothesis that individuals scoring highly on “economic system justification” (ESJ) are less emotionally reactive to stimuli depicting economic suffering. They show that people high in ESJ display lower self-reported emotional and physiological responses to videos depicting homelessness (relative to control videos) than people low in ESJ. They conclude that these results provide evidence that “system justifying beliefs diminish people’s aversion to inequality in the distribution of economic resources”.

This is a very well written paper with an impressive array of methods designed to answer an important question. The use of pre-registration is commendable. I especially appreciated the authors’ clear descriptions of what their studies add to the literature, by highlighting limitations of past studies and how theirs can overcome these limitations. I think this paper has a lot of potential and could be appropriate for Nature Communications. I have a few questions/concerns to address in a revision.

My first and biggest concern has to do with the potential confound between the order of video presentation and content of videos in Studies 3 and 5. I was a little confused by the descriptions of the methods of these studies as there seemed to be some conflicting information between the main text and the supplemental info. So: for Study 3 it appears the homeless video was always presented before the control. This is a big problem because this could produce a spurious finding in the predicted direction, since physiological arousal tends to be higher toward the beginning of a testing session than toward the end. In the main text it seems like only Study 4 counterbalanced the order of the videos (implying that video order Study 5 was not counterbalanced), however the Supplementary Results for Study 5 says “we additionally adjusted for stimulus order in the model”, implying that video order was counterbalanced. So: was the order of videos in Study 5 counterbalanced, or not? If not, this is a problem for the SCR findings because that means there is no evidence to rule out the possibility that the SCR findings are driven by stimulus order (since there is no SCR data from Study 4, which seems to be the only study that counterbalanced the order of videos). Given that much of the novelty of the manuscript hinges on the physiological findings in light of the limits of self-reported emotions pointed out by the authors (and in particular, having multiple converging physiological findings) it would seem important to have at least one dataset that includes SCR findings unpolluted by an order confound.

My second main concern has to do with the broad conclusions the authors want to make about ESJ and emotional responses to “economic inequality”. Ultimately, they have shown that ESJ modulates emotional responses to just one manifestation of economic inequality – videos depicting homeless people. But economic inequality manifests in a number of ways. It would seem critical, to be able to make broad claims about ESJ modulating responses to inequality, that the authors test more than one manifestation of economic inequality. There are a number of ways this could be done – e.g. with videos depicting other aspects of poverty besides homelessness, or with economic games that induce inequality. Demonstrating that ESJ modulates emotional responses to a variety of stimuli related to economic inequality would be very powerful and support the authors’ broad claims. Alternatively, they could tone down the claims in the paper and make them specific to what was tested, i.e., ESJ modulates emotional responses to homelessness (rather than “economic inequality” in general). Of course, this would lessen the overall impact of the claims.

Finally, I found the discussion section to be lacking – only two paragraphs, one summarizing the main findings, and one speculating a bit about implications with few references to the broader literature. A revision should include a more comprehensive discussion section that connects the present findings to the broader literature.

More specific comments:

1. Title references "indifference to the suffering of others" but the paper makes it clear the authors' hypotheses are specific to economic suffering, not suffering in general. So the title seems not to match the more specific arguments and data in the paper.
2. Intro pg. 3 paragraph 1: the authors reference work on advantageous inequality aversion in children and then in the next sentence write "Likewise, chimpanzees have been observed..." referencing work on disadvantageous inequality aversion. This is confusing because the "likewise" suggests the chimpanzee work is also about advantageous inequality aversion.
3. Same paragraph – claims Americans are "unperturbed by widespread economic inequality". Is it that they are "unperturbed", or just unaware? Norton & Ariely have shown that Americans seem to be grossly misinformed about actual inequality of wealth distributions in the US, and so I wondered how much of the purported lack of concern could be explained by this.
4. Several of the studies were preregistered, which is great. It would be helpful, given that the authors include a very large number of regression models in the supplement, if it could be specified exactly which model(s) were preregistered. I think readers will assume the models presented in the results sections are the preregistered ones: if this is the case it would be good if that could be stated explicitly.
5. The presentation of the mediation models could be more clear. A figure (at least in the supplement if not main text) depicting the mediations tested would help.
6. Results page 16: please present the statistics for the tests described in the first paragraph, including the null results for self-reported affect.
7. For discussion: there is work by Fiske and colleagues showing homeless people, as a group, elicit disgust; and separately, work by Inbar & Pizarro suggesting conservatives are more disgust-sensitive. This work might suggest an opposite prediction to that of the authors. The current findings should be discussed in light of this past work.
8. Also for discussion: did the authors have any specific predictions about discrete emotions? The overall results seem rather nonspecific, i.e., ESJ seems to modulate all negatively valenced emotions. It would be helpful if some of the discrete emotions literature could be brought to bear in making sense of these findings.
9. The pre-registration for what appears to be Study 3 recommends 80 participants based on a power calculation. Study 3 ultimately had 42 participants. Should we be concerned that the final sample was half the required size determined by a power calculation? If not, why not?

Responses to Comments Made by Reviewer #1 (Identified as Mina Cikara)

Reviewer #1 wrote that:

The studies are more than competently executed, the results are consistent and robust, and the manuscript is very clearly written. I also want to commend the authors for pre-registering their predictions and making their data available. Finally: THAT TITLE!!! If it isn't already clear, I really enjoyed reading this paper and I think it makes an important contribution to our understanding of how ESJ shapes micro-reactions to macro factors.

We are extremely gratified by the reviewer's response to our work and thank her for providing such thorough and helpful comments.

On p. 3: The discussion jumps from relationships between SJ and fairness to conservatives and life satisfaction/happiness. I understand that L/R and SJ are related but they're not the same so it makes it harder to follow the logic. It's also worth noting that the absence of distress (as predicted by the IPT) does not necessarily require or indicate the presence of high satisfaction/happiness. As such, it might just be cleaner to review literature regarding relative absence of negative affect among high SJs in the intro.

We agree with the reviewer that in the previous version of the manuscript we oscillated between describing the effects of political conservatism and those of economic system justification.

I don't see the EC scale in the robustness check regressions for Experiment 2

The reviewer is correct that the models for robustness checks do not include the EC scale. This is because—following the preregistered analysis plan—we first conducted mixed-effects linear regressions to examine whether ESJ moderates emotional responses to homeless and CF videos. Next, to anticipate concerns about confounding variables, we examined whether these results were robust to inclusion of the BIDR, gender, race, age, religiosity, and income as covariates (Table S35–40). Only after these primary analyses did we test whether our results reflected economic system justification rather than differences in dispositional empathy (Tables 2 and 3). The robustness check models produce similar findings regardless of whether EC is included or not. We have not reported all details of the robustness check models that included EC, however, because of the excessive number of terms in the model, which makes the regression tables extremely large. We would be happy to include these tables, if you wish.

Just to make sure I'm understanding Tables 2 and 3 correctly, each of the interaction columns is that interaction with the main effect of the other scale in the model, correct? E.g., the ESJ X H column indicates the predicted interactions on person-directed and system-directed outcomes treating EC as a covariate?

That is correct. Table 2 shows the predicted ESJ \times Homeless interaction adjusting for EC and the EC \times Homeless interaction, whereas Table 3 shows the ESJ \times CF interaction while adjusting for EC and the EC \times CF interaction.

I think Table 6 has the experiments labeled incorrectly. Should be experiments 3, 5 + IDA (not 2, 4 + IDA).

We thank the reviewer for bringing this typo to our attention. We have fixed it in the revised manuscript.

I have an alternative hypothesis for why the continuous affect data didn't replicate Experiments 1 and 2, which is foreshadowed in my first specific comment above: the continuous measure captured Ps levels of POSITIVE affect (which they registered by sliding a finger left or right on a touchpad). But if you're watching videos about homelessness and CF there ought not be ANY positive affect. If I were a P, I would have kept the dial set at 0 (any other value would have reflected schadenfreude). Ideally Experiments 3-5 would have asked about intensity of negative affect instead.

To clarify, the left-most side of the touchpad was labeled 0 (“extremely negative and unpleasant”) and the right-most side of the touchpad was labeled 100 (“extremely positive and pleasant”). By default, the initial location of the finger on the touchpad was set at 50 (defined as neither negative nor positive in the instructions presented to participants). During stimulus presentation, E-prime software recorded the participants’ finger position on the touchpad every .5 seconds. In hindsight, it might have been easier for participants to respond if we had labeled 0 as neither negative nor positive and included negative values to indicate unpleasant feelings (e.g., -50 to 0). We thank the reviewer for thinking through these procedural and interpretational details with us.

Looks like there is missing text and punctuation here: “Contrary to our preregistered hypothesis, however, we did observe a significant, although smaller, ESJ X CF Video interaction effect on person-directed sadness, system-directed anger, sadness and disgust was marginally significant ESJ X CF Video interaction effect on person pity Notably, the effect of ESJ X Video Type was significantly larger (with some as twice bigger) for the ESJ X Homeless interaction contrast as compared to the ESJ X CF condition on person-directed sadness and system-directed anger, sadness, and disgust.

Again, we thank the reviewer for her careful reading. This error has been fixed in the revised version of the Supplementary Information. The passage now reads as follows:

“Contrary to our preregistered hypothesis, we observed a significant, although smaller, ESJ \times CF Video interaction effect on person-directed sadness as well as system-directed anger, sadness, and disgust. Notably, the effect of ESJ \times Video Type was significantly larger (with some as twice bigger) for the ESJ \times Homeless interaction contrast as compared to the ESJ \times CF condition on person-directed sadness and system-directed anger, sadness, and disgust (see Tables S29–S34).”

Responses to Comments Made by Reviewer #2

Reviewer #2 wrote that:

I wondered whether esj was negatively correlated with negative affect in the homelessness condition. That is, whereas the authors report that the esj x condition interaction was generally significant, and the simple effect of condition was generally stronger at 1sd below the mean in esj than at 1sd above, it was unclear whether the simple slope describing the esj-affect relation in the homelessness condition was significant, as one would expect on the theory that esj “blunts the negative emotional impact of inequality.”

In response to the reviewer’s comment, we have now incorporated these analyses into the Supplementary Information. Here is an overview of the results: In Studies 1 and 2, in the homeless condition the simple effects of ESJ on person-directed sadness and system-directed anger, sadness, and disgust were all significant. In Study 1, the simple effect of ESJ on pity was significant, but in Study 2 this effect was not significant. In Study 3, in the homeless video condition the simple effects of ESJ on corrugator activity and SCL were significant. In Studies 4 and 5, in the homeless video condition, the simple effect of ESJ on *corrugator* activity was not significant. In Study 5, in the homeless video condition the simple effect of ESJ on SCL was marginally significant.

In Study 6, when participants encountered poor target, the simple effects of ESJ on system-directed anger and disgust were indeed significant. However, the simple slope of person-directed anger was not significant. When participants encountered a rich target, the simple effects of ESJ on system-directed anger and disgust and person-directed anger were all significant. When participants encountered a rich target, the simple effect of ESJ on self-directed sadness was marginally significant, but the simple effect of ESJ on jealousy toward the rich target was not significant (see revised SI for statistical details).

I wondered whether the data could be better characterized as showing that low esj heightens negative affect in response to economic suffering, rather than high esj blunting negative affect. This appears to be the case for the result depicted in Figure 1, where individuals higher in esj showed just as much negative affect as all participants did in response to someone who suffered from cystic fibrosis (i.e., even at high esj, participants were well above the midpoint in expressing negative emotion in the homelessness condition, far higher than negative emotion experienced in response to the control video, and on par with negative emotion in response to cystic fibrosis). Individuals low in esj expressed even more negative emotion in response to the homeless target (compared to higher esj participants). Thus, at least for this study, it doesn’t appear that high esj really blunted negative affect, as much as low esj elevated it (if we take the cystic fibrosis video as a reference point for the experience of negative affect).

We believe that the reviewer raises an interesting point here, but it is not one that we can resolve with the data at hand. At this point, we are limited to drawing comparative conclusions between low and high ESJ participants, and we cannot really distinguish

between high ESJ “blunting” negative affect and low ESJ “elevating” it. We will consider this interesting question when designing studies in the future.

A more minor empirical note: given such high reliabilities for the person and system composites (e.g., table 3), it seems more parsimonious to report results with composites alone in the main text, perhaps leaving results with individual components to supplemental materials for interested readers.

We appreciate this suggestion and have discussed it extensively amongst ourselves. The consensus we have arrived at is that having more detailed information about discrete emotions is a real strength of this research program, so we wish to report the results separately for specific emotions, rather than emphasizing results based on composite measures, which gloss over potentially important differences that have been identified in research on discrete emotions.

On a theoretical level, I wondered whether these data are as consistent with social dominance theory as they are with system justification theory. To be clear, I completely agree that these data are consistent with system justification theory – that is, it is plausible that some combination of epistemic, existential, and relational motives would lead to general and economic system justification, which in turn could reduce negative affect in response to poverty. It seems plausible in addition, or alternatively, that individuals who are favorable to social stratification would likewise justify the current economic system, with consequences for affective responses to poverty. An exploratory test of this could be done with the data from experiment 3, where social dominance orientation (sdo) was measured. That is, just as analyses were done to examine if esj mediated the effect of conservatism on affect, one could examine if general system justification (also measured in exp. 3) and sdo simultaneously predict esj, controlling for one another, with downstream effects on affect.

We thank the Reviewer for this suggestion. In the first physiological experiment (Study 3), we did indeed measure SDO (see SI) but found that it did not predict physiological responses to homelessness. Therefore, we have added the following footnote on p. 14.

In Study 3, we also measured Social Dominance Orientation (SDO; see SI) and ran regression models predicting physiological responses on the basis of SDO, Video Condition (homeless vs. control), and their interactions. None of the coefficients for the interactive effect of SDO and video condition were significantly different from zero in these models.

Responses to Comments Made by Reviewer #3

Reviewer #3 wrote:

This is a very well written paper with an impressive array of methods designed to answer an important question. The use of pre-registration is commendable. I especially appreciated the authors' clear descriptions of what their studies add to the literature, by

highlighting limitations of past studies and how theirs can overcome these limitations. I think this paper has a lot of potential and could be appropriate for Nature Communications.

We are extremely gratified by this response and thank the reviewer for his/her generosity in sharing these reactions with us.

My first and biggest concern has to do with the potential confound between the order of video presentation and content of videos in Studies 3 and 5. I was a little confused by the descriptions of the methods of these studies as there seemed to be some conflicting information between the main text and the supplemental info. So: for Study 3 it appears the homeless video was always presented before the control. This is a big problem because this could produce a spurious finding in the predicted direction, since physiological arousal tends to be higher toward the beginning of a testing session than toward the end. In the main text it seems like only Study 4 counterbalanced the order of the videos (implying that video order Study 5 was not counterbalanced), however the Supplementary Results for Study 5 says “we additionally adjusted for stimulus order in the model”, implying that video order was counterbalanced. So: was the order of videos in Study 5 counter-balanced, or not? If not, this is a problem for the SCR findings because that means there is no evidence to rule out the possibility that the SCR findings are driven by stimulus order (since there is no SCR data from Study 4, which seems to be the only study that counterbalanced the order of videos). Given that much of the novelty of the manuscript hinges on the physiological findings in light of the limits of self-reported emotions pointed out by the authors (and in particular, having multiple converging physiological findings) it would seem important to have at least one dataset that includes SCR findings unpolluted by an order confound.

As noted above, we take full responsibility for not making these details clearer in the initial submission. In fact, video order was counterbalanced in Study 5 (as it was in Studies 1, 2, and 4). We have clarified this in the revised manuscript, pointing out that the order was fixed only in Study 3 (the first physiological experiment), but it was fixed in a manner that was opposite to what s/he had guessed; that is, the control video was always presented before the homeless video. Therefore, our results cannot be explained in terms of habituation processes. We thank the reviewer for bringing this issue to our attention, and we thank you for providing us with the opportunity to correct the misunderstanding.

My second main concern has to do with the broad conclusions the authors want to make about ESJ and emotional responses to “economic inequality”. Ultimately, they have shown that ESJ modulates emotional responses to just one manifestation of economic inequality – videos depicting homeless people. But economic inequality manifests in a number of ways. It would seem critical, to be able to make broad claims about ESJ modulating responses to inequality, that the authors test more than one manifestation of economic inequality. There are a number of ways this could be done – e.g. with videos depicting other aspects of poverty besides homelessness, or with economic games that induce inequality. Demonstrating that ESJ modulates emotional responses to a variety of stimuli related to economic inequality would be very powerful and support the authors’

broad claims. Alternatively, they could tone down the claims in the paper and make them specific to what was tested, i.e., ESJ modulates emotional responses to homelessness (rather than “economic inequality” in general). Of course, this would lessen the overall impact of the claims.

We thank the reviewer for raising this point, which we regard as extremely insightful. It is true that while homeless people cue economic inequality, so do people who are conspicuously wealthy. Following his/her advice (and your own), we have therefore added a sixth study, which makes use of “daily diary” experience-sampling methods, in which we compare emotional reactions to encounters with rich and poor people in everyday life as a function of economic system justification. Because of the inclusion of this study, we have also added a new co-author, Dr. Ruthie Pliskin of Leiden University, who was instrumental in designing and implementing Study 6.

I found the discussion section to be lacking – only two paragraphs, one summarizing the main findings, and one speculating a bit about implications with few references to the broader literature. A revision should include a more comprehensive discussion section that connects the present findings to the broader literature.

In response to this comment, we have now expanded the discussion section to connect our findings to the broader literature in political psychology, focusing especially on questions about the nature of ideological differences in negativity bias and disgust sensitivity (including references to work by Fiske et al. and Inbar et al., as suggested by the reviewer below). We also address, in a slightly more speculative vein, potential links between emotion regulation mechanisms and the palliative effects of ideological processes.

Title references “indifference to the suffering of others” but the paper makes it clear the authors’ hypotheses are specific to economic suffering, not suffering in general. So the title seems not to match the more specific arguments and data in the paper.

We appreciate the reviewer’s suggestion and, in light of Study 6, which addresses economic inequality in the absence of suffering (i.e., exposure to conspicuous wealth), we have decided to change the title to “Atlas Shrugs: Economic System Justification and Indifference to Inequality.”

Intro pg. 3 paragraph 1: the authors reference work on advantageous inequality aversion in children and then in the next sentence write “Likewise, chimpanzees have been observed...” referencing work on disadvantageous inequality aversion. This is confusing because the “likewise” suggests the chimpanzee work is also about advantageous inequality aversion.

Here we are referring to work by Brosnan, Talbot, Ahgren, Lambeth, and Schapiro (2010), who do find evidence of aversion to advantageous inequality in chimpanzees. Brosnan et al. (2010) noted that “chimpanzees were more likely to refuse a high-value grape when their partner got a lower-value carrot than when their partner also received a

grape. This is quite interesting in light of the current debate in the literature regarding the role of prosocial preferences in primates' behavior.”

Same paragraph – claims Americans are “unperturbed by widespread economic inequality”. Is it that they are “unperturbed”, or just unaware? Norton & Ariely have shown that Americans seem to be grossly misinformed about actual inequality of wealth distributions in the US, and so I wondered how much of the purported lack of concern could be explained by this.

Our reading of the public opinion literature suggests that the relative lack of concern about economic inequality in the U.S. reflects both cognitive processes (such as lack of information, as emphasized by Norton & Ariely) and motivated processes associated with political ideology and economic system justification (as shown, for instance, by Bartels). Therefore, we cite both contributions and have modified the sentence so that it reads as follows: “However, public opinion data suggest that a large percentage of Americans either pay little attention to or are otherwise unperturbed by widespread economic inequality.”

Several of the studies were preregistered, which is great. It would be helpful, given that the authors include a very large number of regression models in the supplement, if it could be specified exactly which model(s) were preregistered. I think readers will assume the models presented in the results sections are the preregistered ones: if this is the case it would be good if that could be stated explicitly.

Of the studies reported in the current manuscript, the exact hypotheses, design, sample size, procedure, data transformation, and exclusion criteria of Studies 2, 4, and 5 were preregistered (see <https://osf.io/2qn2z/registrations>). For these studies, the robustness checks were not preregistered but were included to ensure that our findings were not driven solely by demographic variables (e.g., SES) that are associated with ESJ.

In Study 2, after conducting the preregistered analyses, we also adjusted for empathic concern by including it and its interactions with the video contrasts in a series of additional regression models. These models were not preregistered but were included to rule out the possibility that our findings reflected ideological differences in dispositional empathy in general, rather than economic system justification *per se*. We now seek to make all of this clearer in the text and the SI.

The presentation of the mediation models could be more clear. A figure (at least in the supplement if not main text) depicting the mediations tested would help.

As requested by the reviewer, we have now added mediation figures to the SI.

Results page 16: please present the statistics for the tests described in the first paragraph, including the null results for self-reported affect.

We have now added these details to Table 6.

For discussion: there is work by Fiske and colleagues showing homeless people, as a group, elicit disgust; and separately, work by Inbar & Pizarro suggesting conservatives are more disgust sensitive. This work might suggest an opposite prediction to that of the authors. The current findings should be discussed in light of this past work.”

We thank the author for these suggestions, and (as noted above) we have expanded the general discussion to make greater contact with other research in this area, including the work cited by the reviewer.

Also for discussion: did the authors have any specific predictions about discrete emotions? The overall results seem rather nonspecific, i.e., ESJ seems to modulate all negatively valenced emotions. It would be helpful if some of the discrete emotions literature could be brought to bear in making sense of these findings.

We agree with the reviewer’s observations and recommendations here and have therefore added the following passage about discrete emotions to the general discussion on p. 24:

“In the self-report studies we adopted a discrete emotion approach, because exposure to manifestations of inequality may elicit different negative emotions based on core appraisals associated with them and their target (e.g., the system vs. individuals). Our predictions, nonetheless, focused on how ideology shapes the overall differences in experienced negativity as modulated by economic ideology, when encountering inequality. These differential affective experiences could, in turn, relate to broader ideological differences in subjective well-being as in the broaden-and-build theory of emotions, according to which emotional valence is linked to subjective well-being. Future research could expand on this by tying ideological differences in emotional experiences to longer term subjective well-being and identifying discrete emotions that are involved in this process.”

The preregistration for what appears to be Study 3 recommends 80 participants based on a power calculation. Study 3 ultimately had 42 participants. Should we be concerned that the final sample was half the required size determined by a power calculation? If not, why not?

The Reviewer is correct that in Study 4 (not Study 3, which was not preregistered) our sample size was smaller than what we had planned. This is because, halfway through data collection and at the end of academic term, we noticed that a signal filter setting on one of our BIOPAC modules had been changed erroneously, resulting in missing EDA—and, therefore, skin conductance level—data. (This is also why Study 4 lacks EDA measurements.) Because we had already planned to conduct Study 5, we decided to stop running Study 4, correct the settings on the BIOPAC module, and begin collecting new data for Study 5 at the beginning of the next semester.

Despite being underpowered, the findings from Study 4 are valid and we see no reason to exclude them. Given that data collection was stopped because we discovered a case of

equipment failure—and not, for instance, because we had inspected the data or engaged in other questionable research practices—our risk of Type I error (false positives) should not be inflated (Simmons, Nelson, & Simonsohn, 2011). Therefore, we do not believe the small sample size of Study 4 is a cause for concern, but we would be willing to exclude the study if for some reason you feel that it is.

Once again, we thank you and the reviewers for providing us with extraordinarily helpful feedback, and we hope you agree that the revision is much improved as a result of the comments and suggestions made by you and the reviewers.

Reviewers' Comments:

Reviewer #1:

Remarks to the Author:

The authors have adequately addressed all of my comments/questions.

Mina Cikara

Reviewer #2:

Remarks to the Author:

After reviewing the revised manuscript, my general opinion is that some of the inconsistencies in the central finding of interest (the relationship between esj and affect in the economic inequality condition) should be described a bit further, in order to help readers understand why they should or should not be of concern. First, I should note that the reason I draw attention to this simple slope as the effect of interest is because the ideological palliation thesis most centrally rests on this relation – the esj by condition interaction, which the main text focuses on, only supports the palliation thesis to the extent that high esj Ps do not experience the same level of negative affect as lower esj Ps, in the inequality condition. If the interaction is driven by a different pattern (e.g., a positive esj-affect slope in the control condition and no esj-affect relation in the inequality condition) it could still be the case that the inequality minus control difference in negative affect is greater among low (vs. high) esj Ps, but this would not be consistent with the ideological palliation thesis (if affect does not differ as a function of ideology, in the inequality condition).

If one accepts this premise—that the ideological palliation thesis depends primarily on the esj-affect slope in the inequality condition—it follows that additional explanation is needed to help understand inconsistent findings across dvs and across studies. In the three studies with physiological measures, it appears that the esj-corrugator slope was significant in one study but not significant in the other two. The esj-scl slope, measured in two studies, appeared to be significant in one and marginally significant in the other. The esj and levator relation in the homeless condition was not reported, presumably because the video and esj interaction was not significant. In the self-reported discrete emotion studies, it doesn't appear that the esj – person-directed-empathy relation in the homeless condition is reported for study 1 (though it's significant in study 2). The esj and person-directed sadness relation does appear to be significantly negative in both studies, but the esj and person-directed pity relation is significant in one study but not in the other. By comparison, the esj – system-directed emotion relations (anger, sadness, and disgust), were more consistent. Thus it appears that overall, esj's relation with physiological measures and with person-directed-emotions in the homeless (inequality) condition is not very consistent (and thus not very supportive of the ideological palliation thesis), though esj's relation with system-directed emotions is relatively more consistent.

Additionally, related to the simple slopes just described, it appears these slopes may have derived from regression analyses that were different than the ones presented in the main text (e.g., the simple slopes for study 2 appear to be derived from the regressions presented in tables s29-34, which do not control for empathic concern, whereas the interaction effects presented in the main text in table 2 control for empathic concern. This might not make a difference, but that would be useful to confirm.

I previously noted that it would be helpful to have a figure for each interaction effect of interest, perhaps with all simple slopes/effects reported, and based on the observations above, I continue to think this may be helpful (for fully understanding the pattern of results).

Related to the authors' point that information about discrete emotions is a strength of this research, perhaps this point could be bolstered if there is some elaboration (perhaps in the supplementals) about how discrete emotions are informative. Without such context, it is difficult for researchers outside the field of emotion research to fully appreciate this. For example, it would be helpful to understand if the downstream implications of person-directed sadness, pity, and empathy are expected to differ (which would justify the importance of understanding each discrete emotion).

I appreciate that the authors conducted the additional analysis with SDO in study 3, which helps address the SDT-SJT distinction I raised.

Reviewer #3:

Remarks to the Author:

All of my comments have been satisfactorily addressed. The addition of the experience sampling is especially powerful and strengthens the paper considerably. Congratulations on a nice paper!

Responses to Reviewer 2's Feedback

Reviewer 2:

“[T]he ESJ by condition interaction ... only supports the palliation thesis to the extent that high-ESJ participants do not experience the same level of negative affect as low-ESJ participants in the inequality condition. If the interaction is driven by a different pattern (e.g., a positive ESJ–affect slope in the control condition and no ESJ–affect relation in the inequality condition), ... [t]his would not be consistent with the ideological palliation thesis.”

“In the three studies with physiological measures, ... the ESJ-corrugator slope was significant in one study but not significant in the other two ... [I]t appears that, overall, ESJ's relation with physiological measures and with person-directed emotions in the homeless (inequality) condition is not very consistent (and thus not very supportive of the ideological palliation thesis), though ESJ's relation with system-directed emotions is more consistent.”

Our response:

Reviewer 2 suggests that ideological palliation entails a simple effect of ESJ on affect when inequality is salient (i.e., in the homeless condition). We respectfully disagree on this point. In our view, the simple effect of ESJ in the inequality condition is not critical to testing the palliation thesis.

We define palliation as what happens (or, more accurately, what doesn't happen) when individuals are confronted with situations of inequality. We therefore understand palliation to reflect a pattern in which persons who are low (vs. high) in ESJ experience more negative affect in response to economic disadvantage or advantage. This operational definition led us to preregister only the ESJ–condition interaction and the simple effect of video condition at low ESJ.

We sought to avoid yoking ideological palliation to the simple effect of ESJ in the homeless condition. Our reasoning was that this pattern, though consistent with palliation, would only be detectible *to the extent that there are no baseline differences in affect between high- and low-ESJ participants*. But there is good reason to believe that such differences do exist. For instance, there is evidence that ESJ is especially high among political conservatives and that conservatives tend to display higher levels of negative affect “at baseline” (e.g., Hibbing, Smith, & Alford, 2014; Inbar, Pizarro, Bloom, & Haven, 2009; Oxley et al., 2008; Jost et al., 2017). Indeed, we see hints of just such ideology-related differences in our control

conditions in both the self-report studies (see, e.g., Figures S1–S5) and the physiological studies (see Figure 1A and 1B). Although these differences rarely reached statistical significance, they are consistent with the existing literature and may interfere with our ability to detect simple effects of ESJ in the homeless condition.

Concerns about baseline differences in affect are especially pronounced in the physiological studies. The EMG and EDA literatures emphasize the importance of within-subject effects, as physiological indices are notoriously subject to systematic and unsystematic sources of individual variation (e.g., Boucsein, 2012; Hess, 2009). Thus, we thought it critical to ground our tests of the ideological palliation thesis in predictions about within-subject effects—specifically, the impact of the homeless vs. control manipulation on participants’ affective responses.

In the self-report studies, the ESJ \times condition interaction and the simple effect of ESJ in the homeless condition was significant for two of the three person-directed emotions retained across Studies 1 and 2 (see Tables S14 and S16) and all three retained system-directed emotions (see Tables S17–S19). (Note that in Study 2 we only attempted to replicate findings that were significant in Study 1.)

As Reviewer 2 notes, the simple effect of ESJ in the homeless condition was not as stable in the physiological studies. We suspect that this is due to the theoretical and methodological factors outlined above—a point which we elaborate in the Studies 3–5 Supplementary Discussion in the SI. It is worth noting, however, that simple-slope tests derived from the Integrative Data Analysis—which combines data across physiological studies and thus provides the most statistical power—do reveal significant ESJ–*corrugator* and ESJ–SCL associations in the homeless condition alone.

In sum, we would argue that the simple effect of ESJ in the homeless condition is actually rather consistent across our self-report and physiological studies, although we do not regard this as a necessary consequence of the palliation thesis given the likely presence of other individual difference factors. We hope that this allays Reviewer 2’s concerns about the manner in which we tested the palliation thesis.

Reviewer 2:

[Slopes reflecting the effect of ESJ in the homeless condition and reported in the SI] may have been derived from regression analyses different than the ones presented in the main text (e.g., the simple slopes for Study 2 appear to be derived from the regressions presented in Tables S29–S34,

which do not control for empathic concern, whereas the interaction effects presented in ... Table 2 control for empathic concern.”

Our response:

Reviewer 2 is correct that the simple effects reported in the SI for Study 2 are derived from regressions that do not control for empathic concern (EC)—and not the regression reported in Table 2, which includes EC. We thank the Reviewer for highlighting that we failed to report simple-slope results for the regressions adjusting for EC. These analyses are now reported the SI for Study 2.

Reviewer 2:

“[I]t would be helpful to have a figure for each interaction effect of interest, perhaps with all simple slopes/effects reported.”

Our response:

We thank Reviewer 2 for this suggestion. We have now added interaction graphs to the SI (see Figures S2–S19). The caption of each added figure contains relevant statistics, including all simple-effects tests.

Reviewer 2:

“Related to the authors’ point that information about discrete emotions is a strength of this research, perhaps this point could be bolstered if there is some elaboration ... about how discrete emotions are informative.”

Our response:

We appreciate this excellent suggestion, and have elaborated on the importance of distinguishing and measuring discrete emotions in the supplement (see p. 3 and p. 27).

Reviewer 2:

“I appreciate that the authors conducted the additional analysis with SDO in study 3, which helps address the SDT–SJT distinction I raised.”

Our response:

We very much appreciate Reviewer 2's suggestion, which we believe strengthens our case by bringing into relief the distinction between SDO and ESJ.

References

- Boucsein, W. (2012). *Electrodermal activity* (2nd ed.). New York, NY: Springer Science + Business Media.
- Hess, U. (2009). Facial EMG. In E. Harmon-Jones & J. S. Beer (Eds.), *Methods in Social Neuroscience* (pp. 70–91). New York, NY: Guildford Press.
- Hibbing, J. R., Smith, K. B., & Alford, J. R. (2014). Differences in negativity bias underlie variations in political ideology. *Behavioral and Brain Sciences*, *37*, 297–307.
- Inbar, Y., Pizarro, D. A., Bloom, P., & Haven, N. (2009). Conservatives are more easily disgusted than liberals. *Cognition and Emotion*, *23*, 714–726.
- Jost, J. T., Stern, C., Rule, N. O., & Sterling, J. (2017). The politics of fear: Is there an ideological asymmetry in existential motivation?. *Social Cognition*, *35*, 324-353.
- Oxley, D. R., Smith, K. B., Alford, J. R., Hibbing, M. V., Miller, J. L., Scalora, M., ... Hibbing, J. R. (2008). Political attitudes vary with physiological traits. *Science*, *321*, 1667–1670.

Reviewers' Comments:

Reviewer #2:

Remarks to the Author:

I am persuaded by the authors' explanation concerning why within subjects effects across condition may be more meaningful than simple effects of ESJ in the inequality condition. I additionally appreciate the other changes, and have no further comments.